# Multidisciplinary Professionals’ Perceptions of Home-Visit Oral Care for Older Adults in Integrated Community Care: A Focus Group Interview Study

**DOI:** 10.3390/healthcare13182365

**Published:** 2025-09-20

**Authors:** Se-Rim Jo, Bo-Ram Shin, Jong-Hwa Jang

**Affiliations:** 1Department of Public Health Science, Graduate School, Dankook University, Cheonan-si 31116, Republic of Korea; serim@ync.ac.kr (S.-R.J.); shinbo012@dankook.ac.kr (B.-R.S.); 2Department of Dental Hygiene, Yeungnam University College, Daegu-si 42415, Republic of Korea; 3Department of Dental Hygiene, College of Health Science, Dankook University, Cheonan-si 31116, Republic of Korea

**Keywords:** community care, dental hygienist, interprofessional work, home-visit oral health care, multidisciplinary professionals

## Abstract

**Background/Objectives**: With the acceleration of population ageing the need for integrated support in healthcare and caregiving is increasing, and the societal demand for improved service quality is also increasing. This study aims to explore how multidisciplinary professionals perceive the implementation of home-visit oral care (HVOC) within the Integrated Community Care in Older Adults model, in order to inform the design of future integrated oral health programs. **Methods:** The study participants comprised 16 individuals: eight dental hygienists with experience in HVOC and eight multidisciplinary healthcare providers. Focus group interviews were conducted with these participants, and the data were analysed using Colaizzi’s phenomenological method to derive key themes and categories. **Results:** The analysis revealed four main thematic categories: (1) cognitive aspects (understanding of geriatric diseases and families); (2) technical aspects (effective communication and competence in oral care); (3) value-based aspects (empathy, patient-centredness, professional pride); (4) multidisciplinary organisational efforts (establishing interprofessional collaboration systems and integrated platforms). **Conclusions:** HVOC services provided by dental hygienists were found to promote oral health among older adults. To ensure the sustainability and effectiveness of such services, a customised integrated care model based on multidisciplinary collaboration should be established.

## 1. Introduction

The transition into a super-aged society has emerged as a major global public health issue [1], and the increasing proportion of the older population has intensified societal concern about health-related problems [2,3]. Among these, oral health issues in older adults are increasingly being recognised as a serious public health concern [4]. The oral health of older adults is not merely a dental issue but is closely linked to overall systemic health [3]. Poor oral health can lead to various negative outcomes, including diminished chewing function, poor nutritional status, and social isolation [5]. These problems not only reduce the quality of life of older adults but also contribute significantly to the deterioration of their overall health status [4,6].

Owing to physiological, psychological, and social ageing, older adults often find it difficult to address various daily life challenges on their own, making collaboration among multidisciplinary healthcare professionals essential [7,8,9]. In particular, to meet the complex health needs of older adults, a system that provides person-centred, integrated services through cooperation among experts from various fields is required [10]. This cooperation is based on interprofessional work, the importance of which is increasingly emphasised in the context of an ageing society [11,12].

Since 2019, some local governments in South Korea have implemented pilot projects for integrated community care [13]. Since 2023, the Integrated Community Care in Older Adults (ICOA) program has been in operation, offering a variety of services [14]. The core of the ICOA program is to provide integrated care based on collaboration among diverse healthcare professionals, including dentists, dental hygienists, nurses, traditional medicine doctors, pharmacists, and physical therapists [8,9]. This multidisciplinary approach is a key strategy not only for oral health management but also for improving overall health and quality of life [6,12]. However, despite growing recognition of the importance of oral health management for older adults in an ageing society [4], systematic and sustainable programs and an effective operational model are still lacking [15,16].

A clear understanding of both the synergistic effects and limitations of interprofessional collaboration is a critical prerequisite for developing a practical and efficient home-visit oral care (HVOC) model [17]. Therefore, the experiences, collaborative processes, and institutional challenges encountered in HVOC settings as well as the development of interprofessional education curricula for integrated care [18] need to be explored to guide the advancement of oral health management within the integrated care system.

An in-depth analysis of the experiences and perceptions of healthcare providers working in the ICOA field can provide empirical and practical data for designing future customised HVOC programs based on the ICOA model [19]. Therefore, this study aimed to explore how multidisciplinary professionals perceive the implementation of HVOC within the ICOA model in order to inform the design of future integrated oral health programs.

## 2. Materials and Methods

### 2.1. Study Design

This study adopted a phenomenological approach to gain an in-depth understanding of the experiences of multidisciplinary professionals involved in providing HVOC within the ICOA system and to explore the meaning and essence of these experiences [20]. We conducted a qualitative study involving focus group interviews (FGIs) [21]. Although phenomenological studies often rely on individual interviews, FGIs were employed in this study to encourage interactions among multidisciplinary participants, enabling them to elaborate on and contrast their experiences. This approach provided richer insights into the collective and interprofessional aspects of HVOC implementation.

### 2.2. Participants

The study participants were healthcare professionals working in the ICOA system. Sixteen individuals who provided written consent after being informed of the study’s purpose and procedures were selected. The participants were assigned to one of two focus groups with eight individuals in each group, consistent with the recommended size for focus group interviews (6–10 participants per group). The two rounds of interviews facilitated data repetition and saturation, thereby ensuring that the sample size was sufficient to meet the study objectives [20,21]. The sample size was based on purposive sampling and the principle of data saturation, rather than statistical calculation.

The participants were purposively recruited through ICOA-affiliated institutions and local government networks. Recruitment notices were disseminated via professional channels, and individuals who met the inclusion criteria were invited to participate voluntarily. Eligibility screening and invitations were coordinated by program managers to ensure diversity in professional backgrounds and years of experience.

The inclusion criteria were as follows:Individuals with more than 1 year of practical experience in community-based health care or integrated community care services for older adults;Professionals currently working in the healthcare or caregiving field, including dental hygienists, nurses, social workers, caregivers, physicians/dentists, occupational therapists, and physical therapists;Individuals who voluntarily agreed to participate in FGIs after receiving sufficient explanation of the study’s purpose and methods;Individuals with no difficulties in communication (language or cognition) and who were able to participate in a 1–2-h interview.

The exclusion criteria were as follows:Individuals not currently working in healthcare or caregiving, or with less than one year of relevant experience;Individuals with no practical experience related to the study topic (HVOC);Individuals who, despite receiving an explanation of the study, did not provide written consent;Individuals with severe language or cognitive impairment, or health conditions that made participation in the FGIs difficult;Individuals who had already participated in an FGI for the same study (in order to avoid duplicate responses and minimize potential bias from prior exposure to the interview guide).

The first FGI included eight dental hygienists who had more than 1 year of field experience in providing HVOC under the ICOA system. Their clinical dental work experience was as follows: less than 5 years (1 participant, 12.5%), 5–10 years (2 participants, 25%), 10–15 years (2 participants, 25%), and over 15 years (3 participants, 37.5%). Their experience in HVOC was 1 year (1 participant, 12.5%), 2 years (4 participants, 50%), and over 3 years (3 participants, 37.5%). The second FGI included eight professionals with more than 1 year of experience in the same integrated care-based healthcare field. Their occupations were as follows: one Korean medicine doctor, one pharmacist, one home care nurse, one physical therapist, two local government officials (one social worker, one nurse), and two dental hygienists.

### 2.3. Data Collection and Analysis

Data collection via FGIs was conducted between 1 October and 31 October 2023. Each FGI lasted approximately 60 min, during which participants engaged in a facilitated group discussion that was guided using a semi-structured open-ended questionnaire. The main questions focused on participants’ awareness of older adults’ health issues during the provision of HVOC, the necessity of interprofessional collaboration, roles within integrated community care, experiences in oral health care, and strategies for improving collaboration. A detailed interview guide is presented in Table 1. Some of the semi-structured, open-ended questions were as follows: ‘What do you consider to be the most pressing oral health issues among older adults?’ and ‘How do you perceive the role of multidisciplinary collaboration?’ (Appendix A).

The collected data were transcribed and repeatedly reviewed to extract meaningful statements and derive themes. Additional interviews were conducted as necessary until data saturation was reached. Each interview began in a natural and comfortable atmosphere and was gradually expanded into the research topic. With the participants’ consent, interviews were recorded using a smartphone and subsequently transcribed. The transcripts were checked for accuracy through repeated listening and then shared with the participants for member checking. Interviews were scheduled before or after working hours to avoid conflicts with the participants’ availability. When information was insufficient during the analysis phase, supplementary interviews were conducted for clarification. In addition to verbatim transcripts, field notes were taken during the interviews to document non-verbal cues such as gestures, facial expressions, and pauses as well as the emotional tones of participants. These observations were integrated into the phenomenological analysis to support a more comprehensive interpretation of the data.

During FGIs, non-verbal cues (e.g., gestures, facial expressions, and emotional tone) were documented through detailed field notes and incorporated into the coding and theme development.

For data analysis, we adopted the phenomenological research framework using Colaizzi’s method of analysis [22]. First, audio data collected from interviews with the participants were transcribed into text, using the Naver AI program “CLOVA NOTE 2.6.0,” and repeatedly read to grasp the overall meaning and context. Second, meaningful statements related to the research topic were identified and extracted through a thorough review of the transcripts. Third, the extracted statements were rephrased into general expressions to derive core meanings, and similar statements were categorised and organised into comprehensive theme clusters. Fourth, based on these theme clusters, the key themes that emerged from the experiences of HVOC providers within the ICOA system were identified. Fifth, the results of the analysis were reviewed through consultation with a dental hygiene professor experienced in qualitative research, and the participants confirmed the derived themes and essential structures to ensure the study’s credibility and validity.

To minimize researcher bias, data were independently coded by two researchers, reflexive notes were maintained throughout the analytic process, and member checks were conducted to confirm the credibility of the findings. Through this analytical process, this study aimed to gain an in-depth understanding of the experiences of multidisciplinary professionals involved in providing HVOC within the ICOA.

### 2.4. Ethical Considerations

This study adhered to the ethical principles of the Declaration of Helsinki and was conducted with the approval of the Institutional Review Board (IRB) of Dankook University (IRB No: DKU 2023-07-026-007 on 27 September 2023). Before the interviews, participants were fully informed about the purpose and procedures of the study, including the audio recording of the interviews, and provided voluntary written consent. Anonymity and confidentiality were strictly maintained throughout data collection and analysis. All data were stored as audio files and used solely for research purposes. The participants were assured that confidentiality would be protected and that there would be no disadvantages resulting from participation. Additionally, the participants were informed of their right to withdraw from the study at any time. To ensure anonymity, instead of the participants’ names, assigned case numbers were used during the recordings.

## 3. Results

The FGIs exploring the experiences of multidisciplinary professionals involved in providing HVOC within the ICOA system yielded 134 meaningful statements. These statements were categorised into 10 themes, which were then grouped into four overarching theme clusters (Table 2). The final theme clusters were as follows: (1) knowledge, (2) skill, (3) value, and (4) multidisciplinary organisational efforts.

Although no fundamental discrepancies emerged between dental hygienists and other multidisciplinary professionals, dental hygienists tended to emphasize technical aspects of oral health, while other professionals focused more on systemic and organizational challenges, highlighting the complementary nature of their perspectives.

### 3.1. Theme 1: Knowledge

Multidisciplinary professionals emphasised that the core of interprofessional collaboration from a knowledge-based perspective lies in the systematic and regular sharing of expertise related to older adults’ health. In particular, they pointed out the need to enhance the connection between oral health and overall systemic health by comprehensively understanding and sharing knowledge about the complex health characteristics of older adults, such as systemic diseases, polypharmacy, and cognitive decline (Figure 1).

*“There needs to be a system in which professionals from each field regularly share and collaborate on cases involving older adults’ systemic diseases, medications, and oral health conditions.”* (Dental hygienist H)

*“Older adults are often reluctant to open their mouths, so various professionals need to collaborate in creating psychological stability and identifying their needs.”* (Dental hygienist E)

*“I’ve seen firsthand that oral health is directly linked to systemic health issues like pneumonia and diabetes. Even if oral problems are recognised, the lack of dental knowledge limits intervention. I believe integrating home nursing and oral health care would enable more effective care.”* (Nurse 1)

The professionals agreed that such a structure for integrated health information sharing helps complement the expertise of each profession and is essential for enhancing the effectiveness of integrated community care.

### 3.2. Theme 2: Skill

The multidisciplinary professionals emphasised the importance of personalised consultation and communication skills tailored to the diverse physical and cognitive characteristics of older adults. They pointed out the need for collaborative strategies among professionals to increase practical applicability in the field. They also agreed that oral care skills go beyond hygiene maintenance and are closely linked to swallowing, nutrition, and rehabilitation, making interprofessional collaboration essential (Figure 2).

*“Professionals from each field should collaborate to develop and share tailored consultation techniques for different types of older adults to enhance the quality of service.”* (Dental hygienist E)

*“Oral care is directly related to physical function recovery and eating ability. When training for physical activity, oral pain or dental conditions often affect performance. I would like to connect basic oral assessments with oral hygiene needs.”* (Physical therapist)

*“Dietary habits and medication adherence in older adults are closely associated with oral health. In particular, for older individuals with dementia, effective medication management requires not only collaboration with care workers but also the establishment of a multidisciplinary system in which health information, including oral conditions, is shared among professionals.”* (Pharmacist)

*“Interprofessional collaboration requires continuous and integrated professional training so that oral hygiene management can be connected with swallowing, feeding, and systemic health.”* (Dental hygienist A)

The professionals stressed that the integration of technical skills and a structured training system serves as a critical foundation for improving the quality of HVOC services.

### 3.3. Theme 3: Value

From a value-based perspective, the professionals emphasised that an empathy-centred, patient-oriented approach founded on multidisciplinary collaboration promotes emotional stability and self-esteem recovery, ultimately contributing to an improved quality of life. This approach emerged as the core value of interprofessional collaboration and was found to have a positive impact on the professional pride of healthcare providers (Figure 3).

*“Professionals from different fields need to collaborate in guiding patients through successful experiences and providing emotional support based on empathy.”* (Dental hygienist D)

*“It’s very important to sincerely care for patients and to continuously improve professional expertise through Interprofessional Collaboration.”* (Dental hygienist E)

*“Oral issues are risk factors for systemic diseases and impact overall health in older adults. However, owing to the lack of a home-based care system, effective intervention is difficult. Strengthening the roles of other professions is essential, and early detection and prevention through multidisciplinary approaches are especially important.”* (Nurse 2)

The collective insights confirmed that sharing patient-centred values enhances motivation for collaboration and serves as a catalyst for the sustainability of HVOC services.

### 3.4. Theme 4: Multidisciplinary Organisational Efforts

At the organisational level, clarifying role allocation, building data-driven integrated management platforms, and implementing systematic satisfaction and performance evaluation mechanisms emerged as key factors for enhancing the efficiency and sustainability of HVOC services (Figure 4).

*“It’s necessary to establish clear role-sharing and collaboration systems among professionals—for example, Korean medicine doctors can oversee digestive and dementia care, while dental hygienists can support oral health and masticatory function.”* (Korean medicine doctor I)

*“Professionals from each field must collaborate to build a platform for integrated patient data management and institutional improvement.”* (Social worker)

The multidisciplinary professionals recognised that establishing structured collaboration systems and integrated platforms for HVOC is foundational for expanding service provision and informing policy development.

## 4. Discussion

In an ageing society, the need for HVOC within integrated community care is steadily increasing [14]. This study identified key elements to enhance the effectiveness of HVOC through a multidisciplinary approach by targeting individuals with experience in the ICOA program. Because researcher bias is a potential limitation in qualitative research, especially when working within a predefined framework such as ICOA, steps were taken to mitigate this risk. Independent coding, reflexive documentation of preconceptions, and member checks were used to enhance the credibility and trustworthiness of the findings. The analysis revealed four thematic clusters—knowledge, skill, value, and multidisciplinary organisational efforts—highlighting how interprofessional collaboration can contribute practically and be expanded. The core elements identified in this study should be interpreted as reflecting the perceptions and professional judgments of the participating healthcare providers, rather than as absolute or universally established components. Nevertheless, these insights provide valuable guidance for the development of practical and contextually relevant HVOC protocols.

First, in the knowledge domain, the professionals expressed the need for expanded roles and deeper expertise. An integrated understanding and regular sharing of information, beyond dental knowledge, namely systemic conditions, polypharmacy, cognitive and mental health, and family caregiving environments, are essential. Oral health is closely linked to diabetes, cardiovascular disease, and dementia [23,24,25], whereas polypharmacy-induced xerostomia affects chewing and swallowing functions [26]. In care facilities for older adults, specialised oral care provided by dentists can lead to lower recurrence rates of aspiration pneumonia than standard nursing care [27]. Home-visit oral health education can also improve xerostomia and swallowing-related quality of life among older adults in integrated community care [28]. These findings demonstrate that when dentists and dental hygienists contribute their expertise through multidisciplinary collaboration, the health outcomes of older adults improve. Therefore, systematic knowledge sharing through cross-professional education and joint case meetings may help identify individual needs and contribute to improving both systemic and oral health.

Second, in the skill domain, the challenge lies in bridging service gaps between professions. The multidisciplinary professionals agreed that oral health is directly tied to systemic health and that preventive care is vital. However, the lack of interprofessional information-sharing systems was identified as a major limitation. There was consensus on the need to establish an HVOC system centred on dental hygienists, alongside basic training and manuals for other professions. HVOC should be formally recognised as an essential area within integrated care services. Joint development and training of practical skills, such as personalised consultation, oral assessments, oral massage, facial exercises, and swallowing training, were especially emphasised.

Technical integration among dental hygienists, physical therapists, and nurses is a fundamental component of integrated community care, enabling the simultaneous management of physical function and nutritional status of older adults [29]. However, vague role definitions, hierarchy-related conflicts, and inefficient workflows can hinder collaboration between the health and welfare sectors [29]. Communication among multidisciplinary personnel is a key factor influencing organisational cooperation in ICOA [30], and effective communication can enhance service quality and ultimately contribute to improved quality of life among older adults [31,32].

Third, in the value domain, empathy-based, patient-centred approaches were considered the ethical foundation of multidisciplinary collaboration. Alleviating patient anxiety and guiding them toward positive experiences not only improves quality of life but also boosts providers’ professional pride [33]. Several studies have shown that multidisciplinary approaches enhance not only physical health but also psychological stability, social support, and independence [34,35,36,37]. These core elements should be carefully considered when implementing HVOC programs.

Fourth, with regard to organisational efforts, clarifying roles, building integrated platforms, and establishing institutional foundations emerged as core strategies. Successful integrated community care requires local governance built on horizontal public–private networks centred on the community. Strengthening local government delivery systems and securing specialised personnel are critical aspects [19]. A prime example is the Program of All-Inclusive Care for the Elderly (PACE) in the U.S., where multidisciplinary teams deliver care with outstanding outcomes in health maintenance and independent living support [38]. To activate integrated care, building HVOC teams led by dental hygienists and expanding integrated management platforms and infrastructure based on patient data are essential requirements.

In summary, future multidisciplinary HVOC programs should evolve in the following directions: continuous reinforcement of interprofessional collaboration, ongoing education and training to enhance service providers’ skills and communication, and expansion of empathy-based, patient-centred approaches. Policy support and establishment of integrated management platforms are necessary to improve the effectiveness and sustainability of multidisciplinary collaboration.

These findings not only support the results of previous research on the importance of integrated care but also extend the literature by showing how multidisciplinary professionals perceive specific challenges in implementing HVOC within the ICOA framework. The emphasis on collaboration across disciplines challenges conventional divisions between medical and dental services. Furthermore, the contradictory perspectives expressed regarding resource allocation highlight an underexplored tension that warrants further investigation.

### Strengths, Limitations, and Future Research Directions

This study is academically significant as it is, to our knowledge, the first to conduct an in-depth analysis of strategies for multidisciplinary collaboration in HVOC by focusing on ICOA intervention providers. Its findings offer essential insights for the development of tailored HVOC protocols, which are increasingly in demand and based on interprofessional collaboration.

This study has several limitations. First, considering that qualitative research emphasises the representativeness and suitability of participant selection [39], our study sample was limited to healthcare professionals engaged in the ICOA framework, which may restrict the generalizability of the findings to other professional groups. Second, as the study was conducted in a single regional context, regional bias cannot be ruled out. Third, as with most FGIs, the possibility of social desirability bias, in which participants may have adjusted their responses according to perceived social norms or group expectations, cannot be ignored. To minimize this risk, we fostered a non-judgmental atmosphere and conducted member checks to enhance credibility. Thematic clusters were, therefore, interpreted with caution, and steps such as member checking and inclusion of minority opinions were taken to mitigate bias. Finally, dentists were not included in this study because the aim was to explore the perspectives of non-dental professionals commonly involved in community and home-visit care for older adults. This approach provided valuable insights into how oral health is perceived and managed outside the dental setting; however, it also limited perspectives related to dental diagnosis and treatment. Future research should therefore include dentists to provide a more comprehensive multidisciplinary view.

The researchers’ professional background in dental hygiene may have influenced the emphasis placed on technical aspects of oral care during data analysis and interpretation. Reflexive documentation and triangulation were employed to minimise this influence. Because researcher bias is a potential limitation in qualitative research, especially when working within a predefined framework such as ICOA, steps were taken to mitigate this risk. Independent coding, reflexive documentation of preconceptions, and member checks were used to enhance the credibility and trustworthiness of the findings.

When compared with global models such as the U.S. Program of All-Inclusive Care for the Elderly (PACE), the Korean ICOA system demonstrates similar goals of multidisciplinary integration but differs in its healthcare structure and rapidly ageing demographics. These cultural and systemic features may limit the direct generalisability of our findings to other contexts. However, because the ICOA model is embedded within South Korea’s unique healthcare system and demographic context, the transferability of the findings to other integrated care models, such as PACE, may be limited. Our results should therefore be interpreted within the cultural and systemic context of South Korea.

Future studies should expand the scope to include more diverse professionals and regional characteristics. In addition, longitudinal research is also needed to validate the effectiveness of customised HVOC protocols that reflect the core elements identified in this study.

## 5. Conclusions

Involvement of multidisciplinary professionals with experience in ICOA-recognised HVOC programs is a key factor in not only improving the oral health of older adults but also enhancing their overall quality of life. The participants emphasised that interprofessional collaboration in HVOC must be based on the following four aspects: knowledge, skill, value, and multidisciplinary organisational efforts. To activate HVOC within integrated community care, the following requirements must be met: continuous strengthening of multidisciplinary collaboration systems, ongoing training and education to enhance provider competencies, expansion of empathy-based and patient-centred approaches, policy support, and development of integrated management platforms. Through these efforts, a sustainable HVOC system that promotes the health and well-being of older adults can be established.

## Figures and Tables

**Figure 1 healthcare-13-02365-f001:**
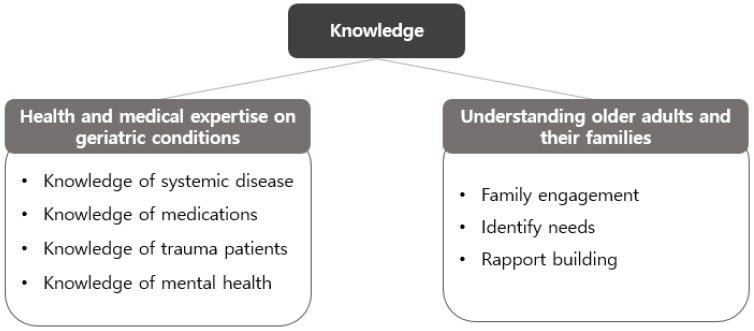
Knowledge domains addressed in this study: (1) Sharing professional expertise in health and medical care related to geriatric conditions; (2) Understanding older adults and their families.

**Figure 2 healthcare-13-02365-f002:**
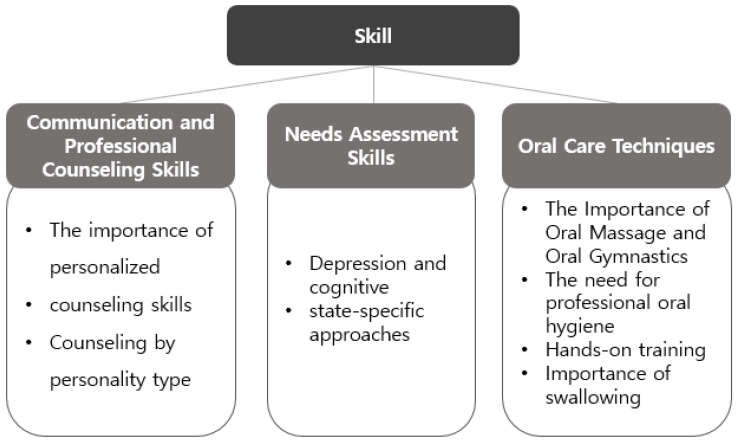
Skill domains highlighted in this study include: (1) Personalized consultation and communication; (2) Professional counseling and needs assessment; (3) Oral care techniques.

**Figure 3 healthcare-13-02365-f003:**
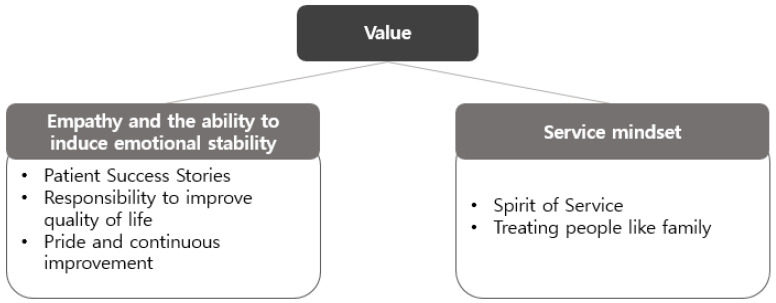
Value domains emphasized in this study: (1) Patient-centered approach with empathy and emotional stability; (2) Service-oriented mindset.

**Figure 4 healthcare-13-02365-f004:**
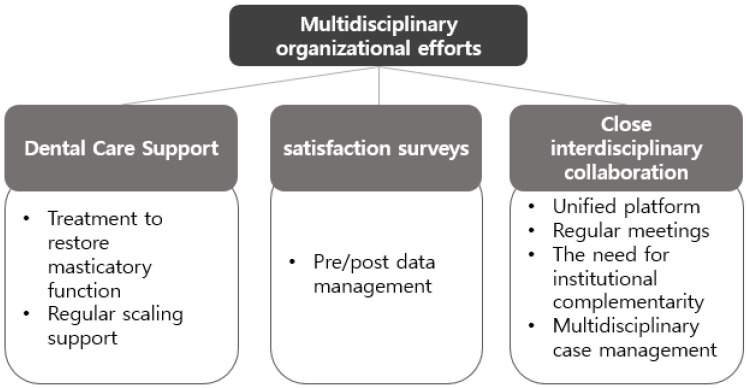
Multidisciplinary organizational efforts in this study are structured into three subdomains: (1) Dental care support; (2) Satisfaction surveys and feedback systems; (3) Close interdisciplinary collaboration.

**Table 1 healthcare-13-02365-t001:** Questionnaire.

Question		Contents
1. Opening questions		-Greeting
2. Introduction question		-The most serious systemic and oral health problems affecting older adults
3. Conversion question		-The need for multidisciplinary practice guidance and alignment in community integrated care
4. Main question	4.1. Care recipients Aspects of care	-Routine assessment items for older adults’ health care (daily, weekly, regularly, or as needed)
		-Key areas and examples of interventions that should be prioritised in older adults’ health management
		-Methods and key focuses of health assessment for older adults in community integrated care
		-Your personal strategies (practical know-how) in the field of geriatric care based on your professional experience
	4.2. Provider collaboration aspects	-Cases requiring multidisciplinary collaboration due to poor oral health
		-Methods of communication in multidisciplinary approaches
		-Goals and focus of multidisciplinary integrated care
		-Strategies to enhance the competencies of community integrated care providers
5. Closing question		-Questions or additional experiences

**Table 2 healthcare-13-02365-t002:** Experiences of multidisciplinary healthcare program providers.

Categories	Division
Knowledge	-Health and medical expertise on geriatric conditions
	-Understanding older adults and their families
Skill	-Communication and professional counselling skills
	-Needs assessment skills
	-Oral care techniques
Value	-Empathy and the ability to induce emotional stability
	-Service mindset
Multidisciplinary Organisational Efforts	-Dental care support
	-Satisfaction surveys
	-Close interdisciplinary collaboration

## Data Availability

The data presented in this study are available upon reasonable request from the corresponding author.

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
