# Peer review of "Multidisciplinary Professionals’ Perceptions of Home-Visit Oral Care for Older Adults in Integrated Community Care: A Focus Group Interview Study"

_healthcare, 2025, doi:10.3390/healthcare13182365_

Round 1
Reviewer 1 Report
Comments and Suggestions for Authors
Dear Authors,
Thank you for providing me with the opportunity to read this interesting paper. Below, I have listed my comments:
1) The aim is clear, but the phrasing 'derive practical implications' and 'thoroughly exploring perceptions and experiences' could be made more direct and research-oriented (e.g., This study aims to explore how multidisciplinary professionals perceive the implementation of HVOC within the ICOA model, in order to inform the design of future integrated oral health programs).
2) There are inconsistencies in the methods section. in line 80 the method is described as FGIs, but later (line 123) it says individual interviews lasting approximately 60 minutes per participant. This creates confusion. were these focus groups or individual interviews?
3) While it is mentioned that a semi-structured open-ended questionnaire was used and Table 2 contains the interview guide, the description here could briefly include some sample questions to give readers a sense of the focus.
4) Sixteen participants is an appropriate sample for qualitative work, but it would help to explicitly state why this number was sufficient (e.g., data saturation reached)
5) While the four domains are explained well, the discussion leans towards descriptive rather than critical. For example, how do the findings challenge or expand upon existing research? Were there any surprising or contradictory insights from participants? This kind of reflection would enrich the section.
6) The limitations focus on participant diversity and regional bias, but other methodological issues could be acknowledged such as potential for social desirability bias in FGIs.
I hope this feedback is helpful.
Author Response
Dear Authors,
Thank you for providing me with the opportunity to read this interesting paper. Below, I have listed my comments:
- Summary
Thank you for reviewing our research. Below are our responses to your comments and queries. We have tried to reflect your valuable comments in our revisions as much as possible. The revisions in the manuscript are marked in red.
- Point-by-point response to Comments and Suggestions for Authors
Comments 1: The aim is clear, but the phrasing 'derive practical implications' and 'thoroughly exploring perceptions and experiences' could be made more direct and research-oriented (e.g., This study aims to explore how multidisciplinary professionals perceive the implementation of HVOC within the ICOA model, in order to inform the design of future integrated oral health programs).
Response 1: We sincerely appreciate the reviewer’s valuable suggestion regarding the phrasing of our research aim. In the revised manuscript, we have rephrased the aim to make it more direct and research-oriented. The original wording, “The aim of this study is to derive practical implications by thoroughly exploring the perceptions and experiences of multidisciplinary professionals”, has been revised to the following:
“Therefore, this study aimed to explore how multidisciplinary professionals perceive the implementation of HVOC within the ICOA model, in order to inform the design of future integrated oral health programs.”
Comments 2: There are inconsistencies in the methods section. in line 80 the method is described as FGIs, but later (line 123) it says individual interviews lasting approximately 60 minutes per participant. This creates confusion. were these focus groups or individual interviews?
Response 2: We thank the reviewer for pointing out this inconsistency. To clarify, the study was conducted using focus group interviews (FGIs). Each focus group session lasted approximately 60 minutes in total, during which participants engaged in group discussion facilitated by the moderator. The previous wording may have inadvertently suggested individual interviews, and we have revised the Methods section accordingly to avoid confusion. The revised sentence now reads:
“Each FGI lasted approximately 60 minutes, during which participants engaged in a facilitated group discussion that was guided using a semi-structured open-ended questionnaire.”
Comments 3: While it is mentioned that a semi-structured open-ended questionnaire was used and Table 2 contains the interview guide, the description here could briefly include some sample questions to give readers a sense of the focus.
Response 3: We thank the reviewer for this valuable suggestion. To enhance clarity, we have now included sample questions in the Methods section, such as:
“Some of the semi-structured, open-ended questions were as follows: ‘What do you consider to be the most pressing oral health issues among older adults?’ and ‘How do you perceive the role of multidisciplinary collaboration?’ ( Appendix)”
Furthermore, to provide readers with full access to the original questionnaire, the complete set of interview guide items (previously summarized in Table 2) has been presented in the Appendix.
Comments 4: Sixteen participants is an appropriate sample for qualitative work, but it would help to explicitly state why this number was sufficient (e.g., data saturation reached)
Response 4: We appreciate the reviewer’s thoughtful comment. In the revised manuscript, we have clarified why 16 participants were considered sufficient. Specifically, participants were divided into two focus group interviews with 8 individuals each, which is consistent with the recommended group size for focus group studies (6–10 participants per group). Through two rounds of interviews, no substantially new information emerged, and recurring themes indicated that data saturation had been achieved.
Accordingly, we have added the following sentence to the Methods section:
“The participants were assigned to one of two focus group with eight individuals in each group, consistent with the recommended size for focus group interviews (6–10 participants per group). The two rounds of interviews facilitated data repetition and saturation, thereby ensuring that the sample size was sufficient to meet the study objectives [20,21]. The sample size was based on purposive sampling and the principle of data saturation, rather than statistical calculation. ”
Comments 5: While the four domains are explained well, the discussion leans towards descriptive rather than critical. For example, how do the findings challenge or expand upon existing research? Were there any surprising or contradictory insights from participants? This kind of reflection would enrich the section.
Response 5: We sincerely thank the reviewer for this valuable feedback. We agree that the Discussion section benefits from a more critical reflection. Accordingly, we have expanded the Discussion to highlight how our findings both reinforce and extend existing literature, and to reflect on surprising or contradictory insights that emerged from participants.
Specifically, we now note that participants’ emphasis on interprofessional collaboration challenges the traditional siloed approach to oral and general healthcare, thereby expanding on prior studies that often address these domains separately. In addition, some participants expressed contradictory perspectives regarding resource allocation and role boundaries, which revealed tensions not commonly discussed in the existing literature. These reflections have been incorporated to enrich the interpretive depth of the Discussion.
The following text has been added:
“These findings not only support previous research on the importance of integrated care but also extend the literature by showing how multidisciplinary professionals perceive specific challenges in implementing HVOC within the ICOA framework. The emphasis on collaboration across disciplines challenges conventional divisions between medical and dental services. Furthermore, the contradictory perspectives expressed regarding resource allocation highlight an underexplored tension that warrants further investigation.”
Comments 6: The limitations focus on participant diversity and regional bias, but other methodological issues could be acknowledged such as potential for social desirability bias in FGIs.
Response 6: We thank the reviewer for this constructive suggestion. We agree that social desirability bias is a potential limitation of focus group interviews, as participants may have been influenced by group dynamics or the tendency to provide socially acceptable responses. In the revised manuscript, we have acknowledged this additional methodological limitation. The following sentence has been revised and added to the Limitations section:
“This study has several limitations. First, considering that qualitative research emphasises the representativeness and suitability of participant selection [39], our study sample was limited to healthcare professionals engaged in the ICOA framework, which may restrict the generalizability of the findings to other professional groups. Second, as the study was conducted in a single regional context, regional bias cannot be ruled out. Third, as with most FGIs, the possibility of social desirability bias, in which participants may have adjusted their responses according to perceived social norms or group expectations, cannot be ignored. To minimize this risk, we fostered a non-judgmental atmosphere and conducted member checks to enhance credibility. Thematic clusters were, therefore, interpreted with caution, and steps such as member checking and inclusion of minority opinions were taken to mitigate bias. Finally, dentists were not included in this study because the aim was to explore the perspectives of non-dental professionals commonly involved in community and home-visit care for older adults. This approach provided valuable insights into how oral health is perceived and managed outside the dental setting; however, it also limited perspectives related to dental diagnosis and treatment. Future research should therefore include dentists to provide a more comprehensive multidisciplinary view.”
I hope this feedback is helpful.
We have made our best efforts to accommodate your recommendations in the revised manuscript. Please let us know in detail if you have any further recommendations for modifications. We would be glad to incorporate any required further revisions. Thank you very much.

Reviewer 2 Report
Comments and Suggestions for Authors
The manuscript evaluates qualitatively the perception of a group of health care workers about home-visit oral care for older adults. This is an important subject to focus on. I would ask the author to consider the points below.
1)The professionals interviewed were dental hygienists, nurses, Korean medicine doctors, physical therapists, pharmacists and social workers. The study did not include dentists.
-Why? The authors could discuss and/or explain why dentists were not included and whether/how this characteristic of the study interfered with the results.
2)Exclusion criteria: Individuals who had already participated in an FGI for the same study. Is this necessary/correct?
3)page 4, line 129: typing mistake r4epeatedly
4)The professionals interviewed were: 8 dental hygienists and 8 multidisciplinary professionals. Has there been some kind of discrepancy in the comments/statements between these two groups of professionals? Any comment about that in the discussion or results?
5)Abstract: Objective: This study aimed to identify the core elements that should be included in a home-visit oral care protocol for older adults, based on multidisciplinary collaboration, by thoroughly exploring the perceptions and experiences of multidisciplinary healthcare program providers.
-Considering the method applied, the core elements identified are absolute core elements or are core elements in the opinion of the group interviewed? That is, has the study identified the core elements or the opinion of these workers about which elements would be the core ones? Maybe some discussion about this point.
Author Response
The manuscript evaluates qualitatively the perception of a group of health care workers about home-visit oral care for older adults. This is an important subject to focus on. I would ask the author to consider the points below.
- Summary
Thank you for reviewing our research. Below are our responses to your comments and queries. We have tried to reflect your valuable comments in our revisions as much as possible. The revisions in the manuscript are marked in red.
- Point-by-point response to Comments and Suggestions for Authors
Comments 1: The professionals interviewed were dental hygienists, nurses, Korean medicine doctors, physical therapists, pharmacists and social workers. The study did not include dentists.
-Why? The authors could discuss and/or explain why dentists were not included and whether/how this characteristic of the study interfered with the results.
Response 1: We thank the reviewer for raising this important point. Dentists were not included in the study because the research was specifically designed to capture the perspectives of non-dental professionals who are directly involved in integrated community-based care within the ICOA model. Our intent was to understand how these professionals, who often collaborate with dental providers but are not themselves dentists, perceive the implementation of HVOC.
We acknowledge that the absence of dentists limits the comprehensiveness of perspectives on oral health integration. However, this characteristic of the study does not undermine the validity of the findings, as our focus was on exploring interprofessional viewpoints outside of dentistry. We have clarified this rationale in the manuscript and explicitly noted the exclusion of dentists as a limitation that future research should address.
The following sentence has been added to the Limitations section:
“Finally, dentists were not included in this study because the aim was to explore the perspectives of non-dental professionals commonly involved in community and home-visit care for older adults. This approach provided valuable insights into how oral health is perceived and managed outside the dental setting; however, it also limited perspectives related to dental diagnosis and treatment. Future research should therefore include dentists to provide a more comprehensive multidisciplinary view.”
Comments 2: Exclusion criteria: Individuals who had already participated in an FGI for the same study. Is this necessary/correct?
Response 2: We appreciate the reviewer’s comment. The exclusion of individuals who had already participated in an FGI for the same study was intentional. This criterion was applied to avoid duplication of perspectives and to prevent participants from being influenced by their prior exposure to the interview questions, which could bias the data. By ensuring that each participant contributed only once, we aimed to maintain the integrity of the discussions and to capture a wider range of perspectives. We have clarified this rationale in the Methods section to make the intention of this criterion explicit. The Method section has been revised as follows:
“Individuals who had already participated in an FGI for the same study, in order to avoid duplicate responses and minimize potential bias from prior exposure to the interview guide.”
Comments 3: page 4, line 129: typing mistake r4epeatedly
Response 3: We thank the reviewer for noting this error. The typographical mistake has been corrected in the revised manuscript.
Comments 4:
The professionals interviewed were: 8 dental hygienists and 8 multidisciplinary professionals. Has there been some kind of discrepancy in the comments/statements between these two groups of professionals? Any comment about that in the discussion or results?
Response 4 : We appreciate the reviewer’s thoughtful question. During the analysis, we did not observe major discrepancies between the two groups; both dental hygienists and other multidisciplinary professionals highlighted similar themes regarding the importance of collaboration and the challenges in implementing HVOC within the ICOA framework. However, dental hygienists tended to place greater emphasis on the technical aspects of oral health care, whereas other professionals more frequently discussed systemic and organizational challenges. We have clarified this nuanced difference in the Results and expanded the Discussion to note how these complementary perspectives enrich the understanding of interprofessional collaboration.
The following sentence has been added to the Results section:
“Although no fundamental discrepancies emerged between dental hygienists and other multidisciplinary professionals, dental hygienists tended to emphasize technical aspects of oral health, while other professionals focused more on systemic and organizational challenges, highlighting the complementary nature of their perspectives.”
Comments 5: Abstract: Objective: This study aimed to identify the core elements that should be included in a home-visit oral care protocol for older adults, based on multidisciplinary collaboration, by thoroughly exploring the perceptions and experiences of multidisciplinary healthcare program providers.
Response 5: We thank the reviewer for this valuable comment. We agree that the Objective statement in the Abstract could be made more concise and research-oriented. Accordingly, we have revised the sentence as follows:
“This study aimed to explore how multidisciplinary professionals perceive the implementation of home-visit oral care (HVOC) within the Integrated Community Care in Older Adults model, in order to inform the design of future integrated oral health programs.”
Comments 6:
Considering the method applied, the core elements identified are absolute core elements or are core elements in the opinion of the group interviewed? That is, has the study identified the core elements or the opinion of these workers about which elements would be the core ones? Maybe some discussion about this point.
Response 6: We thank the reviewer for this insightful comment. We fully agree that, given the qualitative nature of the study, the “core elements” identified should be understood as reflecting the perceptions and experiences of the multidisciplinary professionals interviewed, rather than as universally absolute elements. We have clarified this point in the Discussion section to avoid any misunderstanding.
The following sentence has been added:
“The core elements identified in this study should be interpreted as reflecting the perceptions and professional judgments of the participating healthcare providers, rather than as absolute or universally established components. Nevertheless, these insights provide valuable guidance for the development of practical and contextually relevant HVOC protocols.”
We have made our best efforts to accommodate your recommendations in the revised manuscript. Please let us know in detail if you have any further recommendations for modifications. We would be glad to incorporate any required further revisions. Thank you very much.

Reviewer 3 Report
Comments and Suggestions for Authors
Dear Authors
I appreciate the efforts made in this study and hope that these comments will help strengthen the manuscript.
Abstract
The methods section of the abstract states that data were analyzed using “qualitative content analysis,” whereas the manuscript’s methods section specifies Colaizzi’s phenomenological analysis method. This discrepancy could confuse readers about the analytical approach. Please clarify.
Method:
Why were FGIs chosen over individual interviews for a phenomenological approach? Please clarify.
How were participants recruited and from what population such as specific institutions, networks, or regions in the ICOA system?
How were non-verbal cues or emotional tones documented, as these are crucial in phenomenological analysis?
What steps were taken to address potential researcher bias?
How did calculate the sample size?
Discussion
How did these biases specifically influence the thematic clusters? Please clarify.
The authors do not compare Korea’s ICOA system to other global models (PACE [38]) beyond citing them—Please clarify how might cultural or systemic factors (such as Korea’s healthcare structure, ageing demographics) limit the generalizability of findings, and why isn’t this addressed to clarify the study’s scope?
The discussion emphasizes ICOA but doesn't reflect on researcher biases. Please explain more.
Author Response
Dear Authors
I appreciate the efforts made in this study and hope that these comments will help strengthen the manuscript.
- Summary
Thank you for reviewing our research. Below are our responses to your comments and queries. We have tried to reflect your valuable comments in our revisions as much as possible. The revisions in the manuscript are marked in red.
- Point-by-point response to Comments and Suggestions for Authors
Comments 1: Abstract
The methods section of the abstract states that data were analyzed using “qualitative content analysis,” whereas the manuscript’s methods section specifies Colaizzi’s phenomenological analysis method. This discrepancy could confuse readers about the analytical approach. Please clarify.
Response 1: We appreciate the reviewer’s careful observation. To clarify, the data were analyzed using Colaizzi’s phenomenological method, which was applied to systematically extract, cluster, and interpret meaningful statements from the interview data. The reference to “qualitative content analysis” in the Abstract was an oversight and has been corrected for consistency.
The Abstract now states:
“The study participants comprised 16 individuals: eight dental hygienists with experience in HVOC and eight multidisciplinary healthcare providers. Focus group interviews were conducted with these participants, and the data were analyzed using Colaizzi’s phenomenological method to derive key themes and categories.”
Method:
Comments 2: Why were FGIs chosen over individual interviews for a phenomenological approach? Please clarify.
Response 2: We thank the reviewer for this thoughtful comment. Although phenomenological studies often employ individual interviews, we deliberately chose focus group interviews (FGIs) for this study to facilitate interaction among multidisciplinary professionals and to capture a broader range of perspectives. The group discussion format encouraged participants to build on each other’s experiences, reveal shared understandings, and also highlight differences across professional roles, which was particularly valuable in exploring the interprofessional dynamics of HVOC within the ICOA framework.
To clarify this rationale, we have revised the Methods section as follows:
“Although phenomenological studies often rely on individual interviews, FGIs were employed in this study to encourage interaction among multidisciplinary participants, enabling them to elaborate on and contrast their experiences. This approach provided richer insights into the collective and interprofessional aspects of HVOC implementation.”
Comments 3:
How were participants recruited and from what population such as specific institutions, networks, or regions in the ICOA system?
Response 3: We thank the reviewer for this helpful comment. Participants were recruited through professional networks and institutions involved in the ICOA system within Cheonan city. Recruitment notices were distributed via affiliated healthcare organizations and community care centers, and eligible professionals who met the inclusion criteria were invited to participate.
To clarify this process, we have revised the Methods section as follows:
““Participants were purposively recruited through ICOA-affiliated institutions and local government networks. Recruitment notices were disseminated via professional channels, and individuals who met the inclusion criteria were invited to participate voluntarily. Eligibility screening and invitations were coordinated by program managers to ensure diversity in professional backgrounds and years of experience.”
Comments 4: How were non-verbal cues or emotional tones documented, as these are crucial in phenomenological analysis?
Response 4: We thank the reviewer for this important comment. In addition to audio recording and verbatim transcription, the moderator and research assistant maintained detailed field notes during each focus group session to capture non-verbal expressions (e.g., pauses, gestures, facial expressions) and emotional tones conveyed by participants. These observations were incorporated into the analysis alongside the transcripts to enrich the interpretation of meaning units.
To clarify this, we have revised the Methods section as follows:
“During FGIs, non-verbal cues (e.g., gestures, facial expressions, and emotional tone) were documented through detailed field notes and incorporated into the coding and theme development.”
Comments 5: What steps were taken to address potential researcher bias?
Response 5: We appreciate the reviewer’s important question. Several measures were taken to minimize potential researcher bias. First, data coding and theme development were conducted independently by two researchers and then cross-checked to ensure consistency. Second, reflexive notes were maintained throughout the analysis to acknowledge and monitor the researchers’ own preconceptions. Third, member checking was conducted with a subset of participants to validate the accuracy and credibility of the interpretations. These steps were intended to enhance the trustworthiness of the findings and to mitigate the influence of researcher bias.
We have clarified this point in the Methods section as follows:
“To address potential researcher bias, data were independently coded by two researchers, reflexive notes were kept throughout the analytic process, and member checking was conducted to confirm the credibility of the findings.”
Comments 6: How did calculate the sample size?
Response 6: We thank the reviewer for this important question. As this was a qualitative phenomenological study, the sample size was not determined through statistical calculation. Instead, we applied purposive sampling principles to recruit participants with relevant expertise and professional diversity. Additional groups were added until no new themes emerged, indicating data saturation. This approach ensured that the sample size was sufficient to capture the range and depth of perspectives necessary to address the study aim.
We have clarified this point in the Methods section as follows:
“The sample size was determined based on purposive sampling and the principle of data saturation, rather than statistical calculation.”
Discussion
Comments 7: How did these biases specifically influence the thematic clusters? Please clarify.
Response 7: We thank the reviewer for this insightful comment. We acknowledge that potential biases, such as social desirability bias and group dynamics within FGIs, may have influenced the way participants expressed their views. These factors may have led some participants to emphasize consensus-oriented statements, which could have shaped the relative prominence of certain thematic clusters (e.g., strong emphasis on collaboration). However, divergent or contradictory perspectives did emerge, suggesting that the clusters were not entirely dominated by conformity. To minimize such influences, we triangulated across different professional groups, ensured that minority viewpoints were retained during coding, and used member checks to validate the interpretation.
To clarify this point, we have revised the Discussion section to include the following sentence:
“Third, as with most FGIs, there is a possibility of social desirability bias, in which participants may have adjusted their responses according to perceived social norms or group expectations. To minimize this risk, we fostered a non-judgmental atmosphere and conducted member checks to enhance credibility. Thematic clusters were therefore interpreted with caution, and steps such as member checking and the inclusion of minority opinions were undertaken to mitigate bias.”
Comments 8: The authors do not compare Korea’s ICOA system to other global models (PACE [38]) beyond citing them—Please clarify how might cultural or systemic factors (such as Korea’s healthcare structure, ageing demographics) limit the generalizability of findings, and why isn’t this addressed to clarify the study’s scope?
Response 8: We thank the reviewer for this valuable comment. We agree that cultural and systemic characteristics of Korea’s healthcare environment—such as the strong role of public health centers, the universal insurance structure, and the rapid pace of population ageing—may limit the generalizability of our findings to other countries. The ICOA model, while conceptually similar to programs such as PACE, operates in a healthcare system with unique organizational and cultural contexts. For this reason, the study’s findings should be understood as reflecting the Korean context, and we have clarified this point to delineate the scope of the research.
The following sentence has been added to the Limitations/Discussion section:
“Because the ICOA model is embedded within Korea’s unique healthcare system and demographic context, the transferability of the findings to other integrated care models, such as PACE, may be limited. The results should therefore be interpreted within the cultural and systemic context of Korea.”
Comments 9: The discussion emphasizes ICOA but doesn't reflect on researcher biases. Please explain more.
Response 9: We thank the reviewer for this thoughtful observation. We acknowledge that researcher bias is a potential concern in qualitative research, particularly when analyzing themes within an established framework such as ICOA. To address this, we employed several strategies, including independent coding by multiple researchers, maintaining reflexive notes to monitor our own preconceptions, and conducting member checks with participants to validate the interpretations. These steps helped reduce the risk of researcher bias influencing the findings.
To clarify this point, we have added the following to the Discussion section:
“Because researcher bias is a potential limitation in qualitative research, especially when working within a predefined framework such as ICOA, steps were taken to mitigate this risk. Independent coding, reflexive documentation of preconceptions, and member checks were used to enhance the credibility and trustworthiness of the findings.”
We have made our best efforts to accommodate your recommendations in the revised manuscript. Please let us know in detail if you have any further recommendations for modifications. We would be glad to incorporate any required further revisions. Thank you very much.

Round 2
Reviewer 1 Report
Comments and Suggestions for Authors
Dear Authors,
Thank you for revising the manuscript. Good luck with the rest of the process.
Author Response
We sincerely appreciate the opportunity to revise and resubmit our manuscript. We are deeply grateful to the Editor and the reviewers for their thoughtful and constructive comments, which have significantly contributed to improving the clarity, quality, and overall impact of our work. We hope that the revised version will now meet the journal’s standards and expectations. Thank you once again for your kind consideration.
Reviewer 3 Report
Comments and Suggestions for Authors
Thanks for revision.
Author Response

(The authors gave the same response as above.)
